

**Dramatic changes in atmospheric pollution source contributions for a coastal**
**megacity in North China from 2011 to 2020**
Baoshuang Liu[1,2], Yanyang Wang[1,2], He Meng[3], Qili Dai[1,2], Liuli Diao[1,2], Jianhui
Wu[1,2], Laiyuan Shi[3], Jing Wang[3], Yufen Zhang[1,2*], and Yinchang Feng[1,2*]
[1]State Environmental Protection Key Laboratory of Urban Ambient Air Particulate
Matter Pollution Prevention and Control & Tianjin Key Laboratory of Urban
Transport Emission Research, College of Environmental Science and Engineering,
Nankai University, Tianjin, 300350, China
[2]CMA-NKU Cooperative Laboratory for Atmospheric Environment-Health Research,
Tianjin 300350, China
[3]Qingdao Eco-environment Monitoring Center of Shandong Province, Qingdao,
266003, China
*Correspondence to:* Y.F. Zhang (zhafox@nankai.edu.cn) and Y.C. Feng
(fengyc@nankai.edu.cn)





**Abstract**
Understanding the effectiveness of long-term air pollution regulatory measures is
important for control policy formulation. Efforts have been made using chemical
transport modelling and statistical approaches to evaluate the efficacy of the Clean Air
Action Plan (2013-2017, CAAP) and the Blue Sky Protection Campaign (2018-2020,
BSPC) enacted in China. Changes in air quality due to reduction in emissions can be
masked by meteorology, making it highly challenging to reveal the real effects of
control measures. Knowledge gap still existed with respect to how sources changed
before and after the CAAP and BSPC implemented, respectively, particularly in
coastal area where anthropogenic emissions mixed with additional natural sources
(e.g., marine aerosol). This work applied a machine learning-based meteorological
normalization approach to decouple the meteorological effects from air quality trend
in a coastal city in northern China (Qingdao). Secondly, the relative changes in source
contributions to ambient $PM_{2.5}$ with a ~10-year observation interval (2011-2012, 2016,
and 2019) were also investigated. We discovered that the largest emission reduction
section was likely from coal combustions, as the meteorologically normalized $SO_2$
dropped by ~15.5% per year and dispersion normalized $SO_4^{2-}$ decreased by ~41.5%
for annual average. Change in the meteorologically normalized $NO_2$ was relatively
stable (~1.0% $yr^{-1}$), and $NO_3^-$ changed inappreciable in 2016-2019 but significantly
higher than that prior to the CAAP. Crustal dust decreased remarkably after the CAAP
began. Industrial emissions, for example, steel-related smelting, decreased after 2016
due to the relocation of steelmaking enterprises. Note that vehicle emissions were
increased in importance, as opposed to the other primary sources. Similar to other
mega cities, Qingdao also risks increased ozone pollution that in turns facilitate
secondary particles formation in the future. The policy assessment approaches applied
in this work also work for other places where air quality management is highly in
demand to reduce air pollution.

**Key words:** Air quality; Random forest; Dispersion normalization; Source
apportionment; Coastal megacity



## 1 Introduction

Rapid industrial development and energy consumption in China over the past several decades have resulted in severe air pollution (Dai et al., 2021; Huang et al., 2014; Zhang et al., 2012). Fine particulate matter ($PM_{2.5}$, particles with aerodynamic diameter $\leq 2.5$ μm) is the leading health-risk factor for attributable mortality in China (Cohen et al., 2017). It is well-documented that exposure to $PM_{2.5}$ has been associated with increased mortality (Liu et al., 2021b; Joshi et al., 2021; Vodonos and Schwartz, 2021). The world health organization (WHO) recently set the annual average concentration of $PM_{2.5}$ to 5 μg m$^{-3}$. Most countries or regions are facing great challenge now to meet the guideline since their current $PM_{2.5}$ levels are well above the latest threshold.

To alleviate the severe impact of air pollution on the living environment and public health, the State Council of China released a five-year "Air Pollution Prevention and Control Action Plan" in 2013 (hereinafter the "Clear Air Action Plan, CAAP") (http://www.gov.cn/zwgk/2013-09/12/content_2486773.htm, last access: 29 October 2021). This was followed by the tighter "Three-year Action Plan to the Blue Sky Protection Campaign" (hereinafter the "Blue Sky Protection Campaign, BSPC") in 2018 (http://www.gov.cn/zhengce/content/2018-07/03/content_5303158.htm, last access: 29 October 2021). The executions of these measures significantly improved air quality (Jiang et al., 2021), thus gained appreciable health benefits (Huang et al., 2018). Vu et al. (2019) demonstrated that the control measures requested by the CAAP have tremendously reduced the emissions (after meteorologically normalized pollutants) in $PM_{2.5}$, $PM_{10}$, $NO_2$, $SO_2$, and CO in Beijing from 2013 to 2017 by approximately 34%, 24%, 17%, 68%, and 33%, respectively. Xu et al. (2021) found that by 2020, $PM_{2.5}$ reduction measures avoided 3561 thousand morbidity cases and 24 thousand premature deaths in the Beijing-Tianjin-Hebei region.

Evaluation of the effectiveness of air pollution controls is important for control policy formulation to further improve future air quality (Dai et al., 2020). Many studies have been carried out to evaluate the efficacy of control measures around the world. For example, assessments on short-term control measures were made for the 2008 Olympic Games (Schleicher et al., 2012), 2013 Second Asian Youth Games in Nanjing (Qi et al., 2016), 2014





Asia Pacific Economic Cooperation (Xu et al., 2019b), 2015 Military Parade (Wang et al.,
2017), and 2017 Belt and Road Forum for International Cooperation (Ma et al., 2020), as
well as the 2020 COVID-19 worldwide lockdown (Beloconi et al., 2021; Chen et al., 2020a;
Cucciniello et al., 2022; Shi et al., 2021; Wang et al., 2021a). Medium-term (3–5 years)
evaluations on the validity of control measures have also been examined (Li et al., 2021b; Yu
et al., 2019; Zhang et al., 2019). In contrast, long-term (~10 years) evaluations on controls
were rarely reported (Masiol et al., 2019). The majority of such studies have focused
primarily on the changes in concentrations of criterion air pollutants to qualitatively deduce
the efficacy of source control (Cheng et al., 2019; Lyu et al., 2017; Li et al., 2020; Wang et al.,
2014). For example, Vu et al. (2019) and Liang et al. (2016) applied random forest and non-
parametric methods to normalize the impact of meteorological factors to evaluate the changes
in air pollutant concentrations and the effect of control measures in Beijing and other cities in
China over recent time periods. However, quantitative evaluations of source emissions have
not been common (Gulia et al., 2018), due to the lack of long-term particle composition
monitoring (Hopke et al., 2020) and only a handful of studies quantitatively assessing source
contributions smoothed the disturbance of weather conditions.
Qingdao, as an economically developed coastal megacity in northern China, has suffered
severe air pollution (Bie et al., 2021; Gao et al., 2020;  Li et al., 2017). It has been reported
by Li et al. (2021a) that meteorology plays a critical role in the formation of pollution for this
coastal region. In addition, based on measures taken in accordance with the "CAAP" since
2013 and the "BSPC" since 2018, source interventions such as the relocation and
transformation of businesses and industries from the Old Town to port regions (Liu et al.,
2021a) have been implemented to improve the air quality in Qingdao. Up to now, the air
quality in Qingdao has been greatly improved. However, there is no report to date has
evaluated the effectiveness of these control measures based on a long-term time scale after
these control measures were put into practice, especially for quantitating the changes in
source contributions by smoothing the influences of weather conditions. In view of this, our
work was mainly to evaluate the implementation of control measures utilizing the data of
weather-normalized air pollutants, changes in chemical compositions in $PM_{2.5}$ and source
contributions as well as extra source origins from 2011 to 2020. Findings of this work are



expected to provide the basis for policy development for a coastal megacity in the future.

**2 Materials and methods**
2.1 Study region and sampling site

Qingdao is an economically developed coastal megacity of Shandong province, China

(Fig. S1). The variation of local economic and social developments from 2011 to 2019 were
counted and are shown in Fig. S2. During this period, the local resident population continued
to rise, reaching 9499.8 thousand in 2019. The developed area and the possession of civil
motor vehicles also showed upward tendency, attaining 758.2 $km^2$ and 3062 thousand units in
2019, respectively. The total energy consumption had a maximum of 16891 thousand tons
standard coal in 2012 and maintained comparable levels from 2014 to 2019. The industrial
coal burning capacity above the designated scale and the volume of liquefied petroleum
supply both presented downward trend with values of 10965.7 and 30.2 thousand tons in
2019, respectively. The emissions of sulfur dioxide, nitrogen oxide, and dust basically
showed a downward trend from 2011 to 2019, especially in 2017, and the emissions of these
pollutants remained at relatively low levels after 2017, reflecting that the pollution sources
for these particular contaminants had been effectively controlled in Qingdao.

In this study, in order to evaluate the effectiveness of control measures targeted for

polluted sources in the past decade in Qingdao, ambient $PM_{2.5}$ samples were collected at
urban sites over three time periods during 2011-2012, 2016, and 2019. The 2011-2012
samples were collected before the "CAAP" was enacted in 2013, and the 2016 samples were
collected at the end of the "CAAP", while the 2019 samples were collected during the
middle of the "BSPC" policy period. The sampling plan (detailed in the next section) was
designed to capture changes in the data during these periods, as any changes could reflect
changes in the pollution sources during different stages of China's air pollution control
measures. The sites of Licang and Shinan were sampled in 2011-2012, while five additional
sites, Shibei, Laoshan, Chengyang, Huangdao, and Jiaonan were sampled in 2016 and 2019
(Fig. S1). All collection sites were situated on building rooftops ~10–15 m above ground
level and used to collect ambient $PM_{2.5}$ samples. Further descriptions of the seven sampling
sites are shown in Table S1.



2.2 Sampling and analysis

The sampling periods covered all seasons per year and lasted 41, 56, and 64 days for

2011-2012, 2016, and 2019, respectively (Table S2). Particles were gathered simultaneously
to polypropylene filters and quartz filters loaded to sampling instruments. The details of the
sampling instruments and filters in the different years are listed in Table S3. Samples were
collected for a duration of 22 h from 11:00 to 09:00 of the next day. Field blanks and parallel
samples were synchronously collected at each site. Before sampling, to remove some volatile
compounds and impurities, the quartz and polypropylene filters were baked in an oven at 500
°C and 60 °C for 2 h, respectively. After sampling, all the filters were stored at 4 °C before
gravimetric and chemical analyses were conducted.

Before gravimetric analysis, filter equilibration for 48 h was needed under a constant

temperature (20 ± 1 °C) and humidity (45–55%). All filters were weighed by the
microbalances with a resolution of 1 or 10 μg during different sampling periods; detailed
information is listed in Table S4. To ensure the accuracy, static was eliminated before
weighting and all filters were weighed at least twice to meet error requirements (Table S4).
For chemical analysis, the elements of Na, Mg, Al, Si, K, Ca, Ti, V, Mn, Fe, Ni, Cu, Zn, and
Pb were analyzed in different years. For samples collected in 2011-2012 and 2016,
inductively coupled plasma-mass spectrometer (ICP-MS) was applied to determine these
elements. For samples collected in 2019, inductively coupled plasma-optical emission
spectrometer (ICP-OES) was used to measure all related elements. Water-soluble inorganic
ions of $NO_3^-$, $SO_4^{2-}$, $NH_4^+$, and $Cl^-$ were determined using the ion chromatographs during
different years. The organic carbon (OC) and elemental carbon (EC) of samples during
different years were determined using a thermal/optical carbon analyzer, based on the
IMPROVE (in 2011-2012) and IMPROVE_A (in 2016 and 2019) thermal/optical reflectance
protocol. The detailed instrumental information is listed in Table S5 and analysis procedures
and quality controls are described in Text S1 in the supplemental materials as well as prior
works from Liu et al., 2021a), Huang et al. (2021), Wang et al. (2021b), and Tian et al.,

(2014).

2.3 Random forest (RF) based weather normalization

From 1 January 2015 to 31 December 2020, the hourly concentrations of six air





pollutants ($PM_{2.5}$, $PM_{10}$, $SO_2$, $NO_2$, CO, and $O_3$) at the nine national air quality monitoring
stations in Qingdao were collected from the China National Environmental Monitoring
Network (CNEM) (http://106.37.208.233:20035, last access: 29 October 2021). Data
collected from the nine monitoring stations were averaged to represent the pollution level at
city scale. The explanatory variables including the meteorological variables and time
variables were used to build the RF model and predict the air pollutant concentrations
Hourly surface meteorological data including wind speed, wind direction, temperature,
dewpoint, relative humidity, and pressure recorded at Qingdao Liuting International Airport
were downloaded using the "worldMet" R package (Carslaw, 2017). Time variables
included Unix time (number of seconds since 1 January 1970), Gregorian day (day of the
year), month, week, weekday, and hour of the day. Data were analyzed in RStudio with a
series of packages, and the details of the random forest (RF) model and weather
normalization using the RF model are provided in Vu et al. (2019). The training data set was
comprised of 80% of the whole data, with the rest as testing data. After the RF predictive
model was built for every pollutant, the model was then fed with a new dataset comprised of
time variables same with the original dataset and meteorological variables that resampled
from the whole observation. The prediction process was repeated 500 times to predict the
concentration of a pollutant. The 500 predicted concentrations were then averaged to
calculate the weather normalized concentration. The RF based weather normalization
technique has been extensively used to decouple meteorology from the observed
concentrations, thus can detect interventions in emissions over time (Dai et al., 2020;
Grange et al., 2018, 2019).

2.4 Theil-Sen regression

The Theil-Sen regression technique has been commonly used to explore the long-term

trend of pollutants over years. This approach assumes monotonic linear trends (Masiol et al.,
2019). Its principle is to calculate the slopes of all possible pairs of pollutant concentrations,
select the median value, and give accurate confidence intervals (Munir et al., 2013; Sen,
1968). In this study, the data of air pollutants obtained from RF modelling with weather
normalization was de-seasonalized as the Theil-Sen regression being performed. The Theil-





Sen function is provided by the "openair" R package.
2.5 Dispersion normalization

The concentrations of ambient particles are affected by both local emissions and

meteorological dispersion (Sujatha et al., 2016). Dispersion normalization helps stabilize the
variation of concentrations due to atmospheric dispersion (Sofowote et al., 2021); therefore,
in this study, the contributions of local emissions to particle concentrations were highlighted.
Research suggests that the quantities for particles dispersion can be determined by the
ventilation coefficient (VC) (Kleinman et al., 1976; Iyer and Raj, 2013), which is defined as
the multiplication of mixed layer height (MLH) and the mean wind speed (WS) within the
mixed layer (Eq. (1)). Basing on a VC at a given time interval $i$, the normalized concentration
can be obtained by Eq. (2):
$$VC_i = MLH_i \times WS_i \qquad (1)$$
$$C_{vc,i} = C_i \times \frac{VC_i}{VC_{mean}} \qquad (2)$$
where $VC_i$ (m$^2$ s$^{-1}$) is the ventilation coefficient during period $i$, $VC_{mean}$ (m$^2$ s$^{-1}$) is the mean
VC during the whole study period, and $C_{vc,i}$ (μg m$^{-3}$) and $C_i$ (μg m$^{-3}$) are the normalized and
observed concentrations, respectively. In this study, the dispersion normalization was
conducted for ambient PM$_{2.5}$ and chemical compositions and the resolved source
contributions. The surface wind speed at 10 m was replaced with the mean wind speed
through MLH because of the absence of wind speed at different heights (Dai et al., 2020;
Ding et al., 2021). The 3 h resolution data of MLH and WS was derived from archived
meteorology of the National Ocean and Atmospheric Administration
(https://www.ready.noaa.gov/READYamet.php, last access: 29 October 2021), and the
calculated daily MLH and WS data were used in this study.
2.6 Positive matrix factorization (PMF)

In order to assess the effectiveness of pollution control, source categories and their

contribution were estimated by the PMF method. The PMF decomposes a sample
composition dataset ($X$) into two matrices including source profiles ($F$) and source
contributions ($G$) (Paatero and Tapper, 1994). This principle can be refined as follows:
$$x_{ij} = \sum_{k=1}^{p} g_{ik} f_{kj} + e_{ij} \qquad (3)$$





where $x_{ij}$ is the concentration (μg m$^{-3}$) of the $j$th component from the $i$th sample; $g_{ik}$ means the
contribution (μg m$^{-3}$) of the $k$th source to the $i$th sample; $f_{kj}$ represents the source profile (μg
μg$^{-1}$) of the $j$th component from the $k$th source; $e_{ij}$ is the residual (μg m$^{-3}$) of the $j$th
component of the $i$th sample; and $p$ means the number of sources. In this study, US EPA PMF
v5.0 was applied to carry out source apportionment, and the details in treatment of input data
and method detection limits of chemical compositions are described in Table S6 and Text S2,
respectively.
2.7 Potential source contribution function (PSCF)

We performed PSCF to further investigate the origin of polluted source. First, the 72 h

backward trajectories were calculated at 6 h intervals every day with starting height of 100m
above ground level in Qingdao (36.10° N, 120.32° E), using the Hybrid Single-Particle
Lagrangian Integrated Trajectory (HYSPLIT) model in the GIS-based software of TrajStat
(Liu et al., 2020). The weather data was acquired from Global Data Assimilation System with
horizontal resolution of one-degree latitude-longitude (available at http://www.arl.noaa.gov/,
last access: 29 October 2021). PSCF was then analyzed based on the trajectories added to
source concentrations. The study region was divided into equal-sized grid cells, thus the
number of endpoints for given specific values in every cell could be obtained. According to
pre-set threshold criterion, the PSCF value was the proportion of the number of endpoints
beyond the threshold criterion in each cell. To improve the accuracy of the result, weighted
PSCF was calculated. More details are given in Text S3 of supplementary material.

**3 Results and discussion**
**3.1 Variation characteristics of the air quality**
3.1.1 Trend analysis and annual changes

The annual mean concentration of PM$_{2.5}$ and PM$_{10}$ in Qingdao decreased by 38% and

38% from 51 and 98 μg m$^{-3}$ in 2015 to 32 and 61 μg m$^{-3}$ in 2020, respectively. The annual
mean PM$_{2.5}$/PM$_{10}$ was 0.47 $\pm$ 0.02, with little change from 2015 to 2020, suggesting the
significant impact of coarse particle sources (e.g. dust) in Qingdao. The annual mean levels of
SO$_2$ and NO$_2$ declined by 72% and 8% from 27 and 33 μg m$^{-3}$ in 2015 to 8 and 31 μg m$^{-3}$ in
2020, respectively. The concentration of SO$_2$ showed a significant downward trend, while





that of $NO_2$ changed little, probably indicating that the impact of coal-fired sources was
significantly reduced, while the impact of mobile sources might still be obvious. The annual
mean level of CO decreased by 32% from 0.91 mg m$^{-3}$ in 2015 to 0.62 mg m$^{-3}$ in 2020, while
that of $O_3$ changed little with ranging from 71 to 69 μg m$^{-3}$.

In order to shield the impact of meteorological dispersion, the normalized air quality

parameters were acquired using the RF algorithm under 30-year average (1990–2020)
meteorological conditions. The Theil–Sen trends of air pollutant concentrations after weather
normalization by RF modelling are shown in Fig. 1. The decreasing real trend for air
pollutants except for $O_3$ was found after the weather normalization (Fig. 1), indicating that
the air quality is gradually improving in Qingdao. The trends of the normalized air quality
parameters represent the effects of emission control and, in some cases, associated chemical
processes (Vu et al., 2019).  The Theil-Sen trend analysis of air pollutant concentrations and
$PM_{2.5}/PM_{10}$ and $SO_2/NO_2$ after the weather normalization is shown in Fig. 1. Compared with
other air pollutants, the decline rate of $SO_2$ concentration was the highest (a median of 15.5%
yr$^{-1}$), whereas that of $O_3$ concentration was the lowest (0.2% yr$^{-1}$). Note that the decline rate
of $PM_{2.5}$ concentration (6.0% yr$^{-1}$) was higher than that of $PM_{10}$ concentration (5.6% yr$^{-1}$),
which led to a slight downward trend for $PM_{2.5}/PM_{10}$ (0.6% yr$^{-1}$), indicating that the impact
of coarse particle sources such as dust might be prominent. The decline rate of $SO_2$
concentration was higher than that of $NO_2$ concentration (1.0% yr$^{-1}$), resulting in a higher
$SO_2/NO_2$ decline rate of 15.3% yr$^{-1}$, indicating that the control effect of stationary sources
was better than that of mobile sources (Nirel and Dayan, 2001). It was found that CO
concentration also performed an obvious decreasing trend, with the decreasing rate reaching
5.2% yr$^{-1}$, whereas the downward trend of $O_3$ concentration was not prominent. The
normalized medians of $PM_{2.5}$, $PM_{10}$, $SO_2$, $NO_2$, CO and $O_3$ decreased by 2.8, 5.4, 3.4, 0.3,
42.8, and 0.1 μg m$^{-3}$ yr$^{-1}$, respectively (Table S7).

Figure S3 compares the trends of air pollutants before and after normalization from 2015

to 2020, which are largely different depending on meteorological conditions (Vu et al., 2019).
The annual average concentrations of $PM_{2.5}$, $PM_{10}$, $SO_2$, $NO_2$, CO, and $O_3$ after normalization
were higher than the actual observed concentrations. Compared with 2018, the observed
concentrations of air pollutants in 2019 showed an increase in varying degrees; however, the





increasing values of annual average concentrations for $PM_{2.5}$, $PM_{10}$, $SO_2$, CO, and $O_3$ after
normalization decreased, and even the $NO_2$ concentration after normalization also decreased.
This indicates that the meteorological conditions in 2019 reduced the effect of actual control
to some extent. Up to that point, emission control had resulted in reductions of $PM_{2.5}$, $PM_{10}$,
$SO_2$, $NO_2$, CO, and $O_3$ concentrations by 17.7%, 31.9%, 18.4%, 1.7%, 0.3%, and 0.4% from
2015 to 2020, respectively, highlighting that much work is still needed to ensure the decrease
of $NO_2$ and $O_3$ concentrations in the future.

3.1.2 Changes in the air quality in the two control stages

In order to assess the changes in ambient air quality in Qingdao during different policy

control periods, this study analyzed the changes in air pollutant concentrations during two
stages: the CAAP period (stage 1: 2015-2017) and the BSPC period (stage 2: 2018-2020).
The observed annual mean concentrations for $PM_{2.5}$ and $PM_{10}$ during stage 1 were 45 and 89
$\mu g\ m^{-3}$, respectively (Table S8), and their annual average decline rates were 11.9% and 8.0%
after weather normalization, respectively. Compared with stage 1, the annual average
concentrations of $PM_{2.5}$ and $PM_{10}$ observed in stage 2 were 35 and 71 $\mu g\ m^{-3}$, respectively
(Table S8), and the decline range after normalization was reduced, with the decline rates of
5.3% and 7.0%, respectively (Fig. 2). However, $PM_{2.5}/PM_{10}$ ratios during two stages were
less than 0.5, suggesting that the impact of dust sources might be obvious in the two stages.
Note that the mean observed annual concentration of $SO_2$ was 21 $\mu g\ m^{-3}$ in the stage 1 (Table
S8) and its annual average decline rate reached 25% after normalization (Fig. 2), which was
significantly higher than that of other pollutants. Compared with stage 1, the observed annual
average concentration of $SO_2$ in stage 2 was only 8 $\mu g\ m^{-3}$ (Table S8), and the annual decline
rate of $SO_2$ concentration after normalization still reached 17.1% (Fig. 2), indicating that
Qingdao had achieved remarkable results in the control of coal combustion during the two
stages. The observed annual mean concentrations for $NO_2$ and $O_3$ during stage 1 were 34 and
73 $\mu g\ m^{-3}$, respectively (Table S8), and their annual increasing rates after normalization were
1.5% and 2.8%, respectively (Fig. 2). The observed annual mean concentrations of $NO_2$ and
$O_3$ in stage 2 were 32 and 71 $\mu g\ m^{-3}$, respectively (Table S8), while their annual decline rates
after normalization were only 2.7% and 2.0%, respectively (Fig. 2). This indicates that the





impact of motor vehicles in Qingdao could be greater than expected. Meanwhile, $NO_2$ and
volatile organic compounds emitted from motor vehicles are important precursors for the
formation of $O_3$ (Pugliese et al., 2014; Tsai et al., 2010), which were found to have further
enhanced the $O_3$ concentration in Qingdao's atmosphere. The mean observed annual
concentrations for CO were 0.80 and 0.64 mg m$^{-3}$ in stages 1 and 2, respectively (Table S8),
and the annual average decline rate were 11.4% and 3.2% after normalization, respectively
(Fig. 2), suggesting that there might have been a benefit from the significant control effect of
coal-fired sources.
Diurnal variations of concentrations of air pollutants and $PM_{2.5}/PM_{10}$ and $SO_2/NO_2$
after normalization in the two stages are shown in Fig. S4. The diurnal variation in $PM_{2.5}$
concentration in the two stages was basically the same; however, the concentration of $PM_{2.5}$
in stage 2 was significantly lower than that in stage 1. Diurnal variation of $PM_{10}$
concentration in the two stages was similar to $PM_{2.5}$. The daily variations of $PM_{2.5}/PM_{10}$ in
the two stages were basically the same, and the $PM_{2.5}/PM_{10}$ between 06:00-20:00 in stage 2
was slightly lower than that in stage 1, probably suggesting that the impact of dust increased
slightly during this period.  The diurnal variations of $SO_2$ and CO concentrations during
stages 1 and 2 were generally consistent, whereas their concentrations in stage 2 were
substantially lower than those in stage 1, which might indicate that the control effects of coal
combustion in Qingdao in stage 2 was obvious.  In contrast, the diurnal variations of $NO_2$
concentrations in stages 1 and 2 were basically consistent with the values at each time,
suggesting that the impact of motor vehicles in Qingdao might still be significant, especially
the morning and evening peaks and between 21:00 and 23:00 at night. The daily variations of
$O_3$ concentrations were highly consistent in the two stages, especially between 14:00 and
17:00, $O_3$ pollution was still severe. In general, compared with stage 1, the concentrations of
$PM_{2.5}$, $PM_{10}$, $SO_2$, and CO in stage 2 decreased remarkably at all times, while those of $NO_2$
and $O_3$ remained basically unchanged at all times, indicating that the control effect of coal-
fired sources in Qingdao was significant, whereas the impact of motor vehicles and $O_3$
pollution were more obvious.



### 3.1.3 Changes in air quality after the COVID-19 lockdown

In response to the COVID-19 outbreak, a series of lockdown measures were implemented in China to curb the virus transmission, resulting in a significant decrease in traffic and industrial activities. These limitations provided an opportunity to investigate critical pollution sources that could potentially be better managed in the future to further improve the air quality. In order to explore the changes of air quality in Qingdao during the COVID-19 lockdown period, combined with the specific lockdown situation of Qingdao (http://wsjkw.shandong.gov.cn/ywdt/xwtt/202001/t20200124_3420319.html; http://www.shandong.gov.cn/art/2020/3/7/art_119816_350607.html; last access: 29 October 2021), this study divided the lockdown period into three stages: pre-lockdown (1 to 24 January, 2020), full lockdown (25 January to 7 March, 2020), and partial lockdown (8 to 31 March, 2020). The time series and average values of air pollutant concentrations and $PM_{2.5}/PM_{10}$ and $SO_2/NO_2$ during different lockdown stages and their corresponding periods in 2018 and 2019 are shown in Fig. 3 and Tables S9-S10. According to the weather normalization data, compared with that before the lockdown, the concentrations of $PM_{2.5}$, $PM_{10}$, $SO_2$, $NO_2$, and CO decreased substantially during the full lockdown, among which the concentrations of $PM_{10}$ and $NO_2$ decreased the most (49.5% and 49.0%, respectively), followed by $PM_{2.5}$ (47.8%) (Table S11), which was closely related to the significant decrease in traffic and construction activities during the full lockdown (Collivignarelli et al. 2021; Hong et al. 2021; Wang et al. 2021a). Note that the $O_3$ concentration increased apparently by 50.8% during the full lockdown (Table S11), suggesting that the atmospheric oxidation might be enhanced during this period, similar to the study of Chu et al. (2021), Ding et al., (2021), He et al., (2020), and Le et al. (2020). $PM_{10}$ and $NO_2$ concentrations rebounded significantly during partial lockdown, increasing by 20.3% and 21.1% compared with the full lockdown, respectively, likely due to the increased impacts of traffic activities and related road dust. The concentrations of $PM_{2.5}$, $SO_2$, and CO further decreased during the partial lockdown. The study from Yin et al. (2021) showed that the decrease of $PM_{2.5}$ concentration might be mainly due to the meteorological conditions.

Compared with the same period in 2018, the concentrations of $PM_{2.5}$, $PM_{10}$, $SO_2$, $NO_2$, CO, and $O_3$ decreased obviously during the full lockdown, of which the reduction range of



$SO_2$ concentration was the greatest (39.8%), whereas that of $O_3$ concentration was relatively
lowest (1.8%) (Table S12). Compared with the corresponding period in 2019, the
concentrations of $PM_{2.5}$, $PM_{10}$, $SO_2$, $NO_2$, and CO decreased by 34.5%, 44.8%, 27.0%, 32.6%,
and 22.3% during the full lockdown, respectively, while that of $O_3$ increased by 3.9% (Table
S12). This shows that the COVID-19 lockdown measures led to the marked decrease of the
primary emissions of air pollutants. Meanwhile, the concentrations of particulate matter and
$NO_2$ decreased substantially during the full lockdown. Since there are relatively few
industrial enterprises in urban area of Qingdao, $NO_2$ is mainly emitted from motor-vehicles.
Therefore, this suggested that the control of motor-vehicles under normal conditions should
play an important role in the improvement of air quality in Qingdao.

**3.2 Changes in meteorological conditions and chemical compositions**

In this study, the ventilation coefficient in the same period was used to normalize the

concentrations of chemical compositions in $PM_{2.5}$. After reducing the impacts of
meteorological dispersion, the changes in the concentrations of major chemical compositions
in the different years were analyzed to better reflect the impacts of source emissions (Dai et
al., 2020; Ding et al., 2021). In 2011-2012, 2016, and 2019, the annual average MLHs in
Qingdao were 399, 383, and 414 m, respectively (Fig. S5). However, the average wind speed
in 2016 was significantly higher than that in other years, reaching 3.3 m s$^{-1}$. The ventilation
coefficient showed an increasing trend year by year, from 1292.7 to 1555.4 m s$^{-2}$ (Fig. S5),
suggesting that the atmospheric dispersion conditions in Qingdao were gradually increasing.
The average ventilation coefficient of Qingdao in three years was 1432.6 m s$^{-2}$, and higher
MLH usual corresponds to higher wind speed. Time series of observed concentrations and
normalized concentrations of $PM_{2.5}$ and chemical compositions are shown in Fig. S6. The
observed and normalized concentrations of $PM_{2.5}$ during the whole study period were 93 and
83 µg m$^{-3}$, suggesting that unfavorable meteorological conditions generated approximately 10
µg m$^{-3}$ of growth of $PM_{2.5}$, which was significantly lower than that reported by the study of
Ding et al. (2021) during the COVID-19 lockdown in Tianjin.

The annual changes in the observed and dispersion normalized concentrations and

percentages of main chemical compositions in ambient $PM_{2.5}$ are shown in Fig. 4 and Fig. S7.





From 2011-2012 to 2019, the observed concentrations of $SO_4^{2-}$ showed an obvious downward
trend, from 23.5 to 6.7 μg m$^{-3}$. The trend of concentrations of $SO_4^{2-}$ after dispersion
normalization was consistent with the observed concentrations, and the annual average
decline rate was approximately 41.5% (38.1% in 2016 and 44.8% in 2019) (Table S13),
probably suggesting that the impacts of coal-fired sources in Qingdao has decreased
substantially in recent years. In contrast, the observed concentrations and percentages of $NO_3^-$
increased significantly from 2011-2012 (3.5 μg m$^{-3}$) to 2019 (10.0 μg m$^{-3}$), and $NO_3^-/SO_4^{2-}$
increased from 0.14 to 1.50. After dispersion normalization, the concentrations and
percentages of $NO_3^-$ changed inappreciable in 2016-2019 but significantly higher than that
prior to the CAAP. It has been found that ambient $NO_3^-$ in urban mainly originates from the
secondary conversion of $NOx$ emitted by motor-vehicles (Alexander et al., 2020; Liu et al.,
2017; Meng et al., 2008), thereby indicating that the impacts of motor-vehicles in Qingdao
might become increasingly obvious. The observed and normalized concentrations and
percentages of OC and EC basically performed a downward trend from 2011 to 2019. The
OC concentration decreased significantly, and the observed and normalized concentrations
decreased from 13.1 to 7.6 μg m$^{-3}$ and 12.9 to 7.2 μg m$^{-3}$, respectively, which might be related
to the significant decrease in the impacts of coal-fired sources in Qingdao. Note that the
annual variations of observed and normalized concentrations of $NH_4^+$ were consistent with
that of $SO_4^{2-}$, but contrary to that of $NO_3^-$, which might indicate that ammonium mainly
existed in the form of ammonium sulfate and ammonia hydrogen sulfate in Qingdao.

Crustal elements (Si, Al, and Mg) decreased remarkably after the CAAP were in place.

The observed and normalized concentrations of these elements in 2011-2012 were higher
than those in 2016 and 2019, while their concentrations in 2019 were slightly higher than
those in 2016. From 2011-2012 to 2019, the observed concentrations of Si, Al, and Mg
decreased from 10.7 to 1.0 μg m$^{-3}$, 3.1 to 0.5 μg m$^{-3}$, and 1.9 to 0.2 μg m$^{-3}$, respectively, and
the trends of normalized concentrations were consistent with the observed concentrations,
likely suggesting that the impact of dust in 2011-2012 was apparently higher than that in
2016 and 2019, and 2019 rebounded compared with 2016. The trends of the observed and
normalized concentrations and percentages of Ca were consistent. The concentrations and
percentages in 2011-2012 were remarkably higher than that in 2016 and 2019, and the





concentration in 2019 rebounds compared with that in 2016, with the increasing rate of 77.1%
in terms of normalized data (Table S13). This suggests that the impact of construction
activities in 2011-2012 might have been significantly higher than that in 2016 or 2019. The
annual trends of observed and normalized concentrations of Fe were also consistent. The
observed and normalized concentrations in 2011-2012 were 4.0 and 4.6 μg m$^{-3}$, respectively.
After 2016, the concentrations and percentages of Fe decreased substantially, which might be
closely related to the relocation of iron and steel enterprises in Qingdao (Liu et al., 2021a).
The observed and normalized concentrations and percentages of Ni and V basically showed a
downward trend from 2011-2012 to 2019. The concentrations in 2011-2012 were
significantly higher than that in 2016 and 2019, which might indicate that the impact from
ships in 2011-2012 was more obvious. Of course, it might also be related to the impact of
manual dust sources. From 2011-2012 to 2019, the observed and normalized concentrations
and percentages of Na showed a downward trend. The concentration and percentage in 2011-
2012 were significantly higher than those in 2016 and 2019, suggesting that the impact of sea
salt might have decreased in Qingdao in recent years.

**454 3.3 Changes in source contributions**

3.3.1 Source identification
Given that the differences of source profiles during different periods, PMF analysis was
conducted for three data sets corresponding to separate sampling periods (i.e., 2011-2012,
2016, and 2019). The solutions from five to nine factors were examined in terms of scaled
residuals, factor interpretability, and displacement acceptability (Brown et al., 2015; Dai et al.,
2020). An eight-factor solution was chosen as the optimal fits for each data set. The
correlation coefficients (R$^2$) between the observed and calculated concentrations were 0.91,
0.83, and 0.91, respectively (Fig. S8). There were no DISP swaps, and all BS runs had at least
87% agreement with the base case values (Table S14).
The factor profiles estimated from PMF during different periods are shown in Figs. S9-
S11. The first factor was identified as vehicle emissions, because OC and EC both had high
concentrations and explained variations as well as narrow DISP bounds. It is known that the
OC and EC are important tracers for vehicle emissions (Bi et al., 2019; Gao et al., 2016;



Ryou et al., 2018; Xu et al., 2019a). The second factor was characterized by higher
concentration and explained variation of Si, and high Al concentrations, and they all had
narrow DISP ranges. Si and Al were the indicators for fugitive dust (Begum et al., 2011; Jain
et al., 2018; Zhao et al., 2021). The third factor featured relatively high concentrations and
explained variations of OC, $SO_4^{2-}$, and $Cl^-$ with tight DISP intervals. These species were
distinctive tracers for coal combustion (Huang et al., 2017; Song et al., 2021; Tao et al., 2014).
The fourth factor was characterized by high explained variations of Fe and Mn, and
relatively high concentrations of Cu and Zn. Tsai et al. (2020) found that Fe and Mn were
related to basic oxygen, iron ore sintering and steel oxidation refining. Querol et al. (2007)
and Kuo et al. (2007) have reported that Cu and Zn were released from multiple metal
smelting. Therefore, this factor was identified as steel-related smelting. The fifth factor was
dominated by high concentrations and explained variations of $NO_3^-$ and $NH_4^+$ with small
DISP bounds, which was identified as secondary nitrate (Esmaeilirad et al., 2020). It was
found that $SO_4^{2-}$ and $NH_4^+$ presented the highest explained variations and concentrations with
narrow DISP bounds in the sixth factor. Therefore, this factor was assigned as secondary
sulphate (Bove et al., 2016; Jain et al., 2020). The seventh factor was featured by high
concentration and explained variation of Ca with a small DISP bound, which was identified
as construction dust (Zhang et al., 1999; Zhang et al., 2005). The final factor was
characterized by highly explained variations of Na, Ni, and V with narrow DISP intervals. In
addition, the concentrations of Mg, $NO_3^-$, $SO_4^{2-}$, and $Cl^-$ in this factor were also relatively
high. Zhang et al. (2021), Liu et al. (2018), Choi et al. (2013), and Police et al. (2016) have
found that sea salt involves high amounts of Na, Mg, $NO_3^-$, $SO_4^{2-}$, and $Cl^-$. Meanwhile, Ni
and V are the markers of ship emissions (Manousakas et al., 2017; Zong et al., 2018; Xu et al.,
2018). Therefore, this factor was recognized as a mixed source of sea salt and ship emissions.

3.3.2 Change in source contributions
The source apportionment results of ambient $PM_{2.5}$ in Qingdao from 2011-2012 to
2019 are shown in Fig. 5 and Figs. S12-S15. For vehicle emissions, its contribution showed
an increasing trend with each year, from 12.1 μg m$^{-3}$ (7.9%) to 13.6 μg m$^{-3}$ (22.5%). The
contribution of coal combustion performed a significant downward trend, from 21.3 μg m$^{-3}$



(13.9%) in 2011-2012 to 4.5 µg m$^{-3}$ (7.5%) in 2019. The contribution of fugitive dust in 2011-
2012 was up to 35.3 µg m$^{-3}$ (23.1%), significantly higher than 8.5 µg m$^{-3}$ (13.2%) in 2016 and
10.2 µg m$^{-3}$ (16.8%) in 2019, and the contribution in 2019 rebounded compared with 2016.
The contribution of construction dust showed a downward trend year after year, from 14.2 µg
m$^{-3}$ (9.3%) in 2011-2012 to 2.4 µg m$^{-3}$ (4.0%) in 2019. The contribution of steel-related
smelting also showed a downward trend year by year, from 15.9 µg m$^{-3}$ (10.4%) in 2011-
2012 to 3.0 µg m$^{-3}$ (4.9%) in 2019. The significant decline in the impact of steel-related
smelting after 2016 might be closely related to the relocation of iron and steel enterprises in
Qingdao (Liu et al., 2021a). The contribution of secondary nitrate basically performed a
significant upward trend, increasing from 14.5 µg m$^{-3}$ (9.5%) in 2011-2012 to 15.2 µg m$^{-3}$
(25.2%) in 2019, which might be related to the high concentration of precursor ($NO_2$) (Fig. 1)
and the increase in atmospheric oxidation in recent years (Chen et al., 2020b; Fu et al., 2020).
In contrast, the contribution of secondary sulphate showed a significant downward trend,
from 34.2 µg m$^{-3}$ (22.3%) in 2011-2012 to 9.7 µg m$^{-3}$ (16.0%) in 2019, likely due to the
significant decrease in the concentration of its precursor ($SO_2$) (Fig. 1). For sea salt and ship
emissions, the contribution basically performed a downward trend, from 5.7 µg m$^{-3}$ (3.7%) in
2011-2012 to 2.0 µg m$^{-3}$ (3.2%) in 2019.

To shield the impact of meteorology on the source apportionment results, this study used

Eq. (2) to conduct the treatment of dispersion normalization for the source apportionment
results, and then analyzed the annual changes in the contributions of different source
categories, as shown in Fig. S16. The annual changes in the contributions of multiple sources
in Qingdao were basically consistent with the results of direct PMF calculation. The
contribution of vehicle emissions was increasing year by year, and the annual average
increase rate of contribution concentration was 12.1%. However, the contribution of coal
combustion showed a continuous yearly downward trend, with the average annual decline
rate of contribution concentration being 56.8%. For fugitive dust, compared with 2011-2012,
the contribution in 2016 decreased substantially, with a decline rate of contribution
concentration of 68.9%, while it rebounded in 2019, with an increase rate of 25.2%. The
contribution of construction dust performed a continuous yearly downward trend, with the
average annual decline rate of contribution concentration being 55.9%. For the steel-related





smelting, and sea salt and ship emissions, the average annual decline rates of their
contribution concentrations were 55.3% and 46.0%, respectively. In contrast, the contribution
of secondary nitrate showed an increasing trend, and the increase rate of its contribution
concentration was 1.7%, while the contribution proportion increased by more than 70%. The
contribution of secondary sulphate showed a continuous yearly downward trend, and the
average annual decline rate of contribution concentration was 38.7%. Overall, the impacts of
coal combustion and steel-related smelting industrial sources in Qingdao decreased
substantially over the last decade, suggesting that the controlling effects of these sources were
obvious. The impact of motor-vehicles was prominent each year. Qingdao also risks increased
emissions from the increased vehicular population and ozone pollution that facilitate
secondary particles formation in the future. The impact of fugitive dust had decreased in
recent years, whereas its contribution was still obvious. Therefore, the control of motor-
vehicles and dust should be the focus of pollution source control in Qingdao in the future,
while that of coal combustion and industrial sources also should not be ignored.

In this study, the heating season in 2011-2012 was defined from 15 to 29 February,

2012, that in 2016 was defined from 17 to 20 December, 2016, and that in 2019 referred from
12 to 26 January, 2019. The contributions of different sources during different heating
seasons in Qingdao are shown in Figs. S17-S18. Compared with the heating season in 2011-
2012, the contribution of coal combustion decreased significantly in the heating seasons of
2016 and 2019, from 50.2 $\mu g\ m^{-3}$ (31.7%) to 10.6-10.7 $\mu g\ m^{-3}$ (6.4-10.8%). The contribution
percentages after dispersion normalization showed a consistent trend. For vehicle emissions,
the contribution percentages in the heating season increased continuously each year, from 3.9%
in 2022-2012 to 22.3% in 2019. The results after normalization had the same trend,
suggesting that the impact of motor vehicles in heating season was gradually prominent. The
contribution of fugitive dust in the heating season in 2011-2012 (14.2 $\mu g\ m^{-3}$) was
substantially higher than that in 2016 (3.9 $\mu g\ m^{-3}$) and 2019 (12.0 $\mu g\ m^{-3}$). The contribution in
the heating season in 2019 rebounded remarkably compared with that in 2016, and the results
of dispersion normalization were consistent. The contribution of construction dust in the
heating season in 2019 was markedly lower than that in 2011-2012 and 2016. The
contribution of steel-related smelting in the heating season showed a continuous yearly





downward trend, from 22.6 μg m$^{-3}$ in the heating season from 2011-2012 to 4.6 μg m$^{-3}$ in
2019. However, its contribution percentage in the heating season in 2019 was higher than that
in the heating season in 2016, which was consistent with the normalized results, indicating
that the impact of steel-related smelting in the heating season had increased, though the
contribution percentage was low. The contribution of secondary nitrate in heating season in
2016 was up to 61.3 μg m$^{-3}$ (36.3%), which was significantly higher than that of 28.4 μg m$^{-3}$
(28.9%) in 2019 and 16.8 μg m$^{-3}$ (10.6%) in 2011-2012. This was consistent with the results
of the dispersion normalization. It can be seen that although the contribution of secondary
nitrate in the heating season in 2019 was reduced, its contribution was significantly higher
than that of other sources. Similarly, the contribution of secondary sulphate was also higher in
the heating season of 2016 than other years; however, its contribution was clearly lower than
that of secondary nitrate. After dispersion normalization, the contributions of secondary
sulphate basically showed a continuous yearly downward trend. The contribution of sea salt
and ship emissions in the heating season also showed an obvious downward trend, from 10.0
μg m$^{-3}$ (6.3%) in 2011-2012 to 1.4 μg m$^{-3}$ (1.5%) in 2019, and the results after dispersion
normalization were basically consistent. The average decline rate of contribution
concentration was approximately 70%, including 88% in 2016. From this analysis, the
impacts of coal combustion and steel-related smelting in Qingdao were relatively low after
the heating season in 2016, while that of vehicle emissions was prominent each year.
Although the impact of fugitive dust had rebounded in the heating season in 2019, the
contribution was relatively low. The contribution of secondary nitrate in heating season was
substantially higher than that of other sources, and the influence of secondary sulfate
decreased each year. The influence of sea salt and ship emissions in heating season showed a
continuous yearly downward trend.

**3.4 Changes in potential source areas**

Similar to the studies of Liu et al. (2021a) and Dai et al. (2020), according to the source

apportionment results, this study used the PSCF method to analyze the changes in the
potential impact areas of emission sources in Qingdao from 2011-2012 to 2019, and the
results are shown in Fig. 6. For vehicle emissions, the potential impact areas changed greatly





from 2011-2012 to 2019. The potential impact areas in 2011-2012 were located at the
junction of Shandong, Henan, Anhui, and Jiangsu provinces, and the potential impact areas
were mainly located in the south part of Jiangsu in 2016, while in 2019, Tianjin and the
northwest part of Shandong were important impact areas. The potential impact areas for
fugitive dust showed a trend of westward migration from 2011-2012 to 2019. For 2011-2012,
the potential impact areas were located at the junction of Shandong, Henan, Anhui, and
Jiangsu, as well as in the northern part of Shandong. The potential impact areas were located
in the northwestern part of Shandong in 2016, while they were at the junction of Shandong
and Henan in 2019. For coal combustion, the potential impact areas for 2011-2012 were
located at the junction of Shandong, Henan, Anhui, and Jiangsu. In 2016, they moved to the
northwest of Shandong Province and Beijing Tianjin and Hebei region, and the northwest of
Shandong was an important impact area in 2019. For steel-related smelting, Beijing and
Tianjin were the potential impact areas for 2011-2012, while the potential impact area was
located in the Yellow Sea in 2016, which might be related to the relocation of iron and steel
enterprises to a port area in the south of Qingdao (Liu et al., 2021a). This suggests that the air
mass transport in the coastal area could lead the nearby sea areas to become potential impact
areas. The potential impact area in 2019 was mainly located at the junction of Hebei, Henan,
and Shandong.

For secondary nitrate, the potential impact area for 2011-2012 was the junction of

Shandong, Henan, Jiangsu, and Anhui provinces. The potential impact area was mainly
located in the central and southern parts of Shandong in 2016, while two areas were located
in Beijing, Tianjin, and the junction of Hebei, Henan, and Shandong provinces in 2019. For
secondary sulphate, the main potential impact areas for 2011-2012 were located at the
junction of Shandong, Henan, Jiangsu, and Anhui Provinces and the western part of Jilin
Province. The impact of the Middle East of Shandong Province was more obvious in 2016,
while the impact was greater in the south part of Shandong Province, and the junction of
Henan and Jiangsu Provinces in 2019. For construction dust, the main potential impact areas
for 2011-2012 were Beijing, Tianjin, and the western part of Shandong Province, and the
southeastern part of Hebei Province, Shanghai, and the eastern part of Hubei Province in
2016, while the central and western parts of Shandong Province, the junction of Henan and





Shandong Provinces, and the central and southern parts of Anhui Province were the main
impact areas in 2019. For sea salt and ship emissions, the potential impact areas for 2011-
2012 were mainly located in coastal areas of Jiangsu and Shanghai, which were closely
related to the impacts of ship emissions from ports and sea salt in these cities. The Yellow Sea
was the main impact area in 2016 and 2019, and the impact areas in 2019 moved to the south.
Bie et al. (2021) also analyzed the potential impact areas of ship emissions in Qingdao Port
from 2018 to 2019 using the PSCF method, and found that they were mainly located in the
Bohai Bay, Yellow Sea, and Yangtze River Delta. Overall, from 2011-2012 to 2019, the
potential impact areas of different emission sources in Qingdao have changed markedly. In
2019, the potential impact areas for most of the emission sources were mainly located in
Shandong Province and along the border areas between the western or southwest parts of
Shandong and other provinces, while sea salt and ship emissions were mainly affected by
transport on the Yellow Sea.

**4 Conclusions**

A machine learning-based meteorological normalization and a dispersion normalization-

based on ventilation coefficient approaches were applied to decouple the meteorological
deduced variations in air quality time series and multiple source contributions of a coastal
city in northern China (Qingdao), respectively. The concentrations of air pollutants other than
ozone in Qingdao decreased substantially and the air quality improved continuously after the
"CAAP" period, indicating that the control strategies of air pollution in Qingdao over the
years have been proper. The largest emission reduction sections were likely from coal
combustions and industrial emissions from 2011-2012 to 2019, and the decrease of steel-
related smelting after 2016 due to the relocation of iron and steel enterprises. The
contribution of dust in Qingdao decreased remarkably after the "CAAP", but the impact was
still outstanding until 2019. Vehicle emissions were increased in importance, as opposed to
the other primary sources. Qingdao risks increased emissions from the increased vehicular
population and ozone pollution that facilitate secondary particles formation in the future. In
addition, the influence of ship emissions should be gradually reduced. The control of motor-
vehicles and dust should be the focus of pollution source control in Qingdao in the future,





while that of coal combustion and industrial sources cannot be ignored. In addition, the
potential impact areas of different emission sources in Qingdao have changed markedly from
2011-2012 to 2019. The potential impact areas for most of emission sources were mainly
located in Shandong and the border areas between western or southwest Shandong and other
provinces in 2019, while sea salt and ship emissions were mainly affected from the transport
of the Yellow Sea.

**Author contributions**
Baoshuang Liu: Data curation, Writing – original draft, Yanyang Wang: Data curation and
Investigation, He Meng: Data collection, Qili, Dai: Supervision and Review, Liuli Diao: Data
curation, Jianhui Wu: Supervision, Laiyuan Shi: Supervision, Jing Wang: Supervision, Yufen
Zhang: Supervision – review & editing, Yinchang Feng: Supervision – review & editing.
**Competing interests**
The authors declare no competing financial interests.
**Acknowledgements**
The authors are grateful to the Qingdao Eco-environment Monitoring Center of Shandong
Province for collection of particulate matter samples in this study.
**Financial support**
This study was financially supported by the China Postdoctoral Science Foundation (No.
2019M660986), the Tianjin Science and Technology Plan Project (No. PTZWHZ00120) and
the Fundamental Research Funds for the Central Universities: Nankai University (No.

63211074).

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

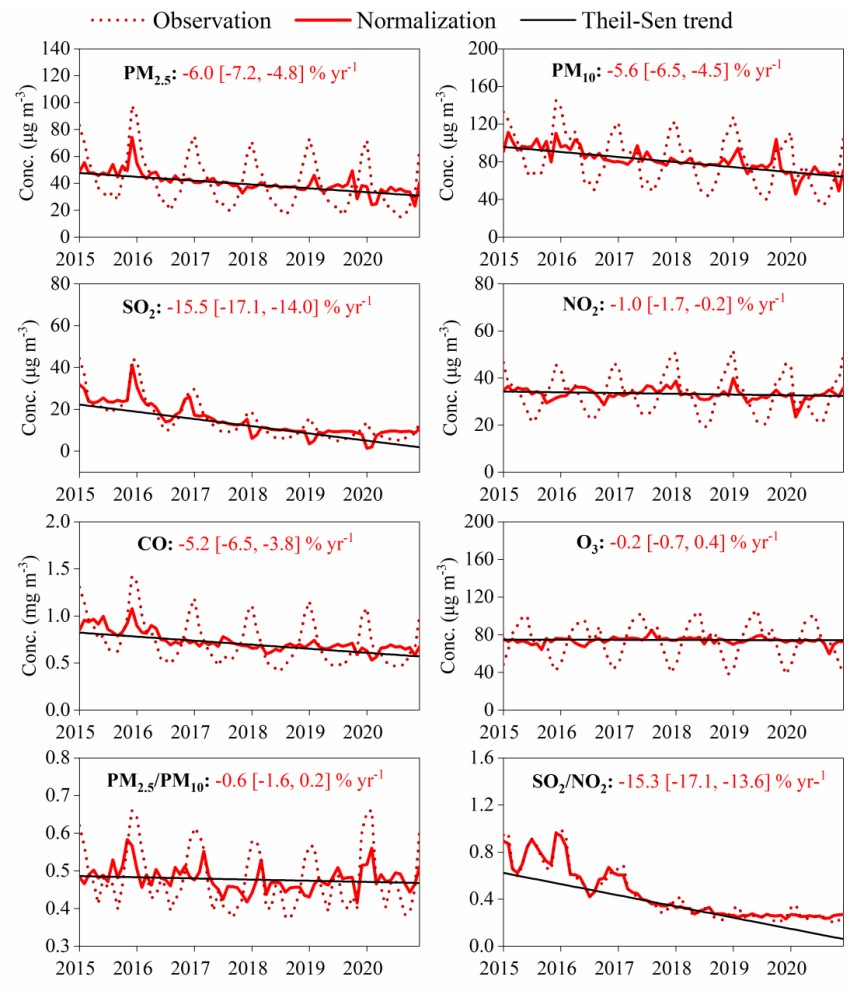


**Figure 1.** Trends of air pollutant concentrations and $PM_{2.5}/PM_{10}$ and $SO_2/NO_2$ from 2015 to 2020. "Observation" represents the observed data, and "Normalization" in represents the modelled concentrations of air pollutants after weather normalization. The black line shows the Theil–Sen trend after weather normalization.

995
996



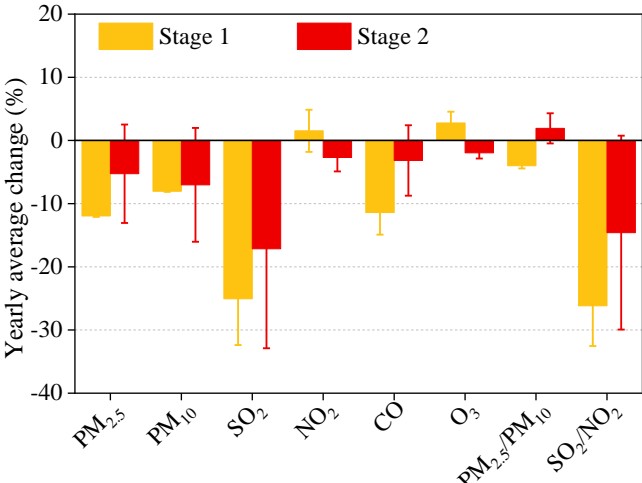

**Figure 2.** Yearly average change of air pollutants and PM$_{2.5}$/PM$_{10}$ and SO$_2$/NO$_2$ during
different pollution-control stages based on the weather normalized data.

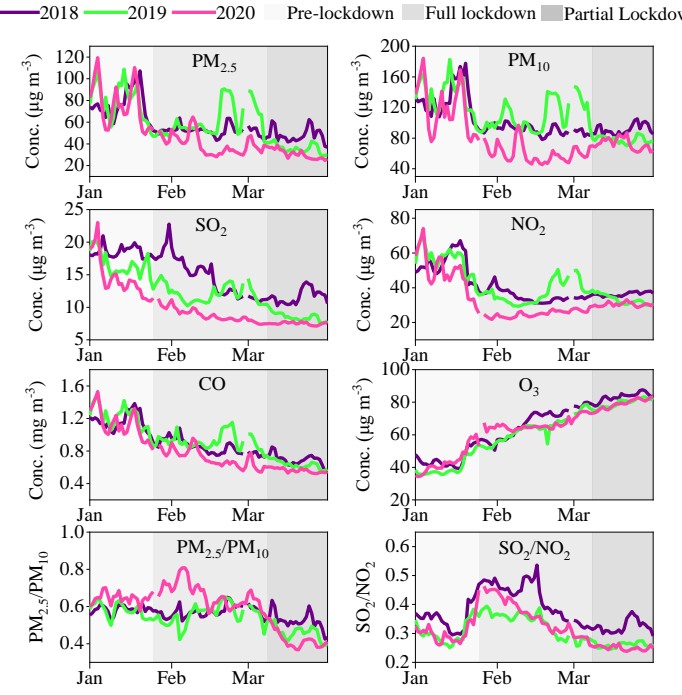

**Figure 3.** Time series of air pollutants concentrations and PM$_{2.5}$/PM$_{10}$ and SO$_2$/NO$_2$ during
the different stages of COVID-19 lockdown start dates or equivalent in 2020 versus 2018 and
2019 based on the weather normalization data.

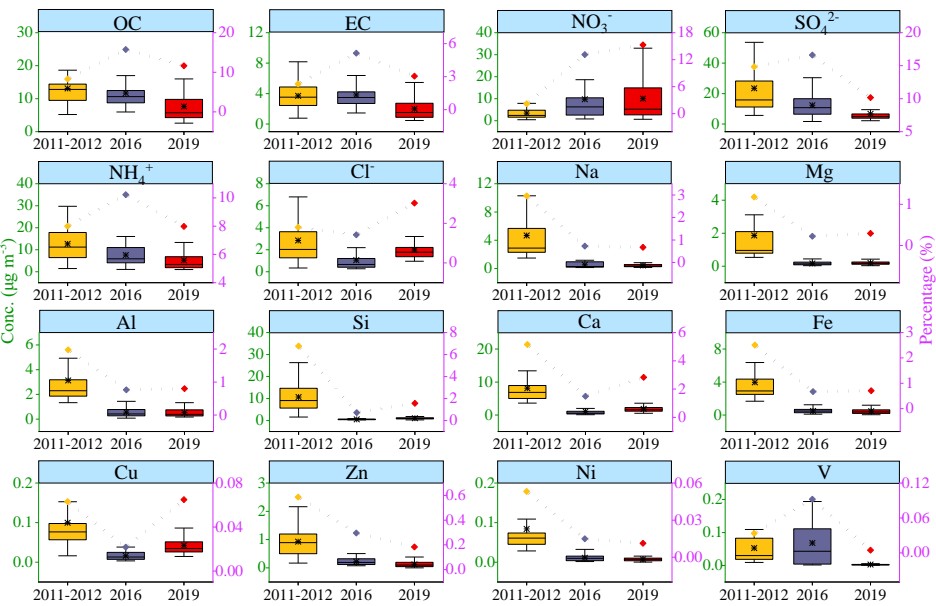

**Figure 4.** Variations of the average concentrations and percentages of major chemical compositions of PM$_{2.5}$ in 2011-2012, 2016, and 2019. Box charts represent concentrations, and line charts represent percentages.

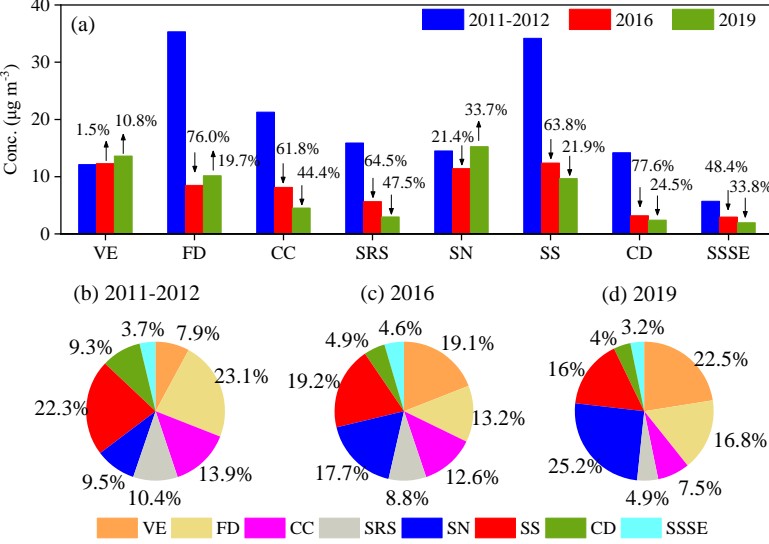

**Figure 5.** Changes in source contributions for 2011-2012, 2016, and 2019. VE represents vehicle emissions, FD represents fugitive dust, CC represents coal combustion, SRS represents steel-related smelting, SN represents secondary nitrate, SS represents secondary sulphate, CD represents construction dust, and SSSE represents sea salt and ship emissions.



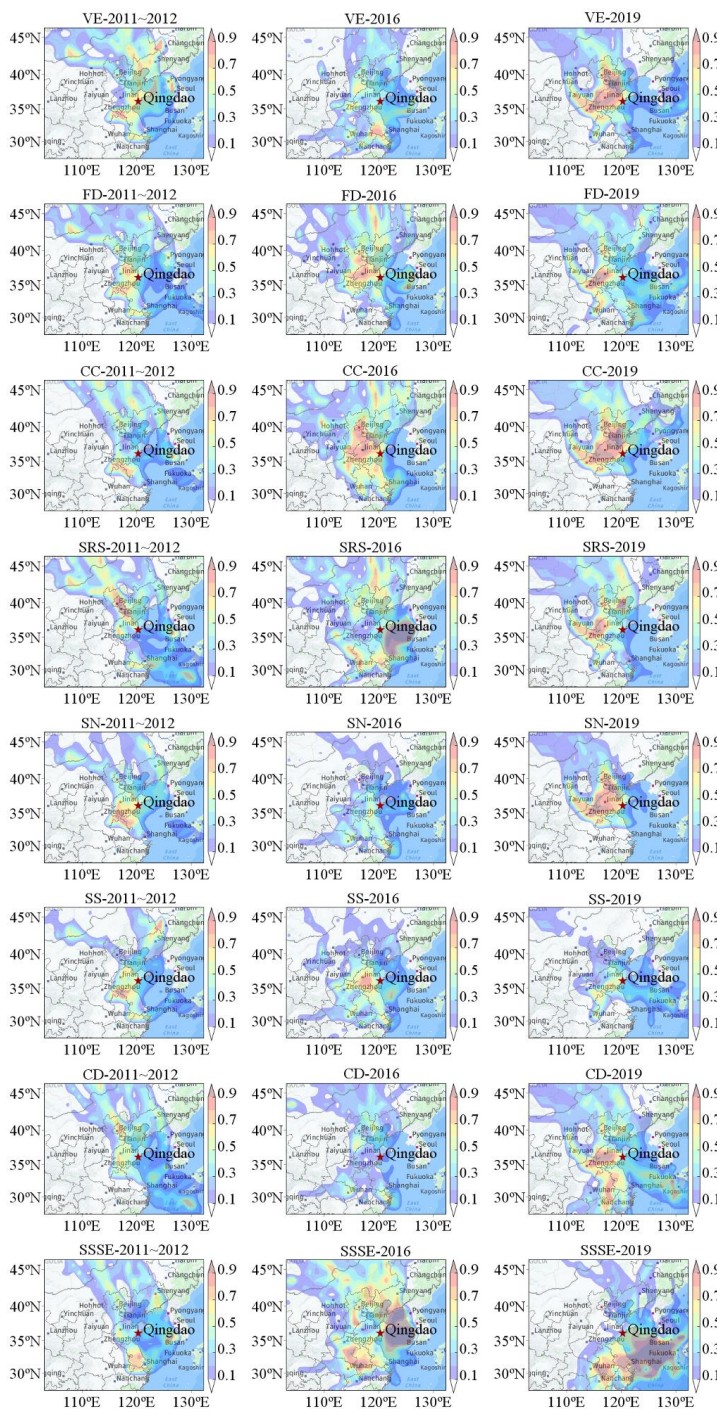


**Figure 6.** WPSCF plots for various emission sources during different periods (base map from Yahoo Maps). VE represents vehicle emissions, FD represents fugitive dust, CC represents





coal combustion, SRS represents steel-related smelting, SN represents secondary nitrate, SS
represents secondary sulphate, CD represents construction dust, and SSSE represents sea salt
and ship emissions.