# Peer review of "Dramatic changes in atmospheric pollution source contributions for a coastal"

_Atmospheric Chemistry and Physics, 2021_

## Author Response (AR1)

**Comments of Anonymous Referee #2**

The manuscript applied a machine learning-based meteorological normalization approach to decouple the meteorological effects from air quality trend in a coastal city in northern China, and further assessed the changes in the contributions of pollution sources in Qingdao in the past ten years. There are some minor issues need to be solved.

1. In the last paragraph of the introduction, if any, it is suggested to add some evaluation articles on the changes in air quality and the effectiveness of control measures before and after major events held in Qingdao (e.g., the 2018 the Shanghai Cooperation Organization summit in Qingdao). Clarify the differences with this paper to support the particularity of this study.

**Response:** Thanks for your advice. We added the related contents in the revised version (on the lines 102-106). The more details as following:

Liu et al. (2020a) assessed the changes in $O_3$ concentrations during the Shanghai Cooperation Organization (SCO) Summit in Qingdao and analyzed the impact of control measures on the emissions reduction of its precursors, and Liu et al. (2020b) also analyzed the reasons for the increase of $O_3$ concentration at nighttime during the SCO Summit.

2. In the sampling and analysis section (L144-145), what is the basis for collecting 22 hours a day? Why not 24 hours?

**Response:** Thanks for your advice. The sampling period was from 11:00 to 09:00 the next day, and the sampling time was 22 hours. The sampling time covers the peak time of morning and evening traffic and the period of strong photochemical reaction at noon, and includes the peaks and troughs of air pollutant concentration. The sampling period of this study is representative. Meanwhile, because the sampling instruments in this study cannot continuously replace the filters, therefore, it needs to take a certain time to replace the sampling filters for subsequent sampling. Furthermore, similar sampling times (22-23 hours) occurred in the studies of Liu et al., (2021), Fang et al. (2017), Turap et al. (2019), and Wang et al. (2016).

Liu, B.S., Wu, J.H., Wang, J., Shi, L.Y., Meng, H., Dai, Q.L., Wang, J., Song, C.B., Zhang, Y.F., Feng, Y.C., Hopke, P.K., 2021. Chemical characteristics and sources of ambient $PM_{2.5}$ in a harbor area: Quantification of health risks to workers from source-specific selected toxic elements. Environ. Pollut., 2021, 268, 115926.

Fang, X.Z., Bi, X.H., Xu, H., Wu, J.H., Zhang, Y.F., Feng, Y.C., 2017. Source apportionment of ambient $PM_{10}$ and $PM_{2.5}$ in Haikou, China. Atmos. Res. 190, 1–9.

Turap, Y., Talifu, D., Wang, X.M., Abulizi, A., Maihemuti, M., Tursun, Y., Ding, X., Aierken, T., Rekefu, S., 2019. Temporal distribution and source apportionment of $PM_{2.5}$ chemical composition in Xinjiang, NW-China. Atmos. Res. 218, 257–268.

Wang, Y.N., Jia, C.H., Tao, J., Zhang, L.M., Liang, X.X., Ma, J.M., Gao, H., Huang, T., Zhang, K., 2016. Chemical characterization and source apportionment of $PM_{2.5}$ in a semi-arid and petrochemical-industrialized city, Northwest China. Sci. Total Environ. 573, 1031–1040.

3. Have the sample data of 2011-2012, 2016, and 2019 been carried out for source analysis, respectively? Please clarify. In addition, the data of all sites were chronologically ordered end to

end for each PMF analysis? PMF is usually used for source analysis based on the long-time data of one site. Please explain the rationality.

**Response:** (1) Yes. The related contents were described in the lines 477-479. (2) Yes. The data of all sites were chronologically ordered end to end for each PMF analysis, and the related contents have shown in text S2 in the supplementary material. (3) In this study, the data of all sampling sites were included in PMF for source analysis, mainly because all sampling sites were located in Qingdao urban area, and the impact source-categories at these sites were the same. The stability of the model running can be improved due to the increase of the number of samples by incorporating multiple sites data into PMF. Escrig et al. (2009) suggested that because the three sampling sites were affected by the same source-categories, therefore, they can be treated as one site and then included in PMF for analysis. Mooibroek et al. (2011) incorporated the receptor data of five sampling sites into PMF for source apportionment, which considered that combining the receptor data of five sites can increase the amount of input data for the model, and then more stable analysis results can be obtained. Furthermore, many related studies have adopted similar methods (Lu et al., 2018; Ezeh et al., 2019; Zhao et al., 2019).

Escrig, A., Monfort, E., Celades, I., Querol, X., Amato, F., Minguillón, M.C., and Hopke, P.K.: Application of optimally scaled target factor analysis for assessing source contribution of ambient $PM_{10}$, J. Air & Waste Manage., 59, 1296–1307, https://doi.org/10.3155/1047-3289.59.11.1296, 2009.

Mooibroek, D., Schaap, M., Weijers, E.P., and Hoogerbrugge, R.: Source apportionment and spatial variability of $PM_{2.5}$ using measurements at five sites in the Netherlands, Atmos. Environ., 45, 4180-4191, https://doi.org/10.1016/j.atmosenv.2011.05.017, 2011.

Lu, Z.J., Liu, Q. Y., Xiong, Y., Huang, F., Zhou, J. B., and Schauer, J.J.: A hybrid source apportionment strategy using positive matrix factorization (PMF) and molecular marker chemical mass balance (MM-CMB) models, Environ Pollut., 238, https://doi.org/10.1016/j.envpol.2018.02.091, 2018.

Ezeh, G, C., Obioh, I, B., Asubiojo, O., Abiye, O, E., and Onyeuwaoma, N, D.: A study of $PM_{2.5-10}$ pollution at three functional receptor sites in a sub-Saharan African megacity, Aerosol Science and Engineering., 3, https://doi.org/10.1007/s41810-019-00044-3, 2019.

Zhao, Z.P., Lv, S., Zhang, Y.H., Zhao, Q.B., Shen, L., Xu, S., Yu, J.Q., Hou, J.W., and Jin, C.Y.: Characteristics and source apportionment of $PM_{2.5}$ in Jiaxing, China, Environ. Sci. Pollut. R., 26, https://doi.org/10.1007/s11356-019-04205-2, 2019.

4. In the process of PMF calculation, the author analyzed DISP and BS (Fig. S9-11, Table S14), but the author did not mention it in the introduction of PMF method. Please add.

**Response:** Thanks for your advice. The related contents have been added in the revised manuscript (on the lines 238-239). The more details as following:

Bootstrap (BS) and displacement (DISP) analyses were used to investigate the effects of measurement error and rotation ambiguity on the resulting solutions.

5. In recent years, scholars have studied the change of air quality in many cities around the world

(such as Beijing), many of which use random forest and other methods. In section 3.11, it is suggested to add the content of comparative analysis with other cities.

**Response:** Thanks for your advice. We added the contents of comparative analysis with other cities in the revised manuscript (on the lines 288-299). The more details as following:

Similar to this study, Vu et al. (2019) found that primary emission controls required by the CAAP in Beijing have led to substantial reductions in $PM_{2.5}$, $PM_{10}$, $NO_2$, $SO_2$, and CO from 2013 to 2017 of approximately 34%, 24%, 17%, 68%, and 33%, respectively, after meteorological normalization. Zhai et al. (2019) suggested that the mean $PM_{2.5}$ decreased across China was 4.6 μg $m^{-3}$ $yr^{-1}$ in the meteorology-corrected data from 2013 to 2018, and the Beijing–Tianjin–Hebei, the Yangtze River Delta, the Pearl River Delta, the Sichuan Basin, and the Fenwei Plain decreased 8.0, 6.3, 2.2, 4.9, and 5.0 μg $m^{-3}$ $yr^{-1}$, respectively. Overall, the concentrations of most air pollutants (i.e., $PM_{2.5}$, $PM_{10}$, $SO_2$, $NO_2$, and CO) in China have showed a decreasing trend in recent years (Zhao et al., 2021; Fan et al., 2020, while that of $O_3$ has performed an increasing trend (Li et al., 2020; Ma et al., 2021), which further facilitated secondary particles formation (Wang et al., 2016; Nøjgaard et al., 2012).

6. L401-404 "The observed and normalized concentrations of $PM_{2.5}$ during the whole study period were 93 and 83 μg $m^{-3}$, suggesting that unfavorable meteorological conditions generated approximately 10 μg $m^{-3}$ of growth of $PM_{2.5}$", can the difference of simple subtraction represent the influence value of meteorology? Is there any basis?

**Response:** Thanks for your advice. In this study, we used the ventilation coefficient to normalize the impact on $PM_{2.5}$ concentrations from the meteorology, and the observed and normalized concentrations of $PM_{2.5}$ during the whole study period were 93 and 83 μg $m^{-3}$, respectively, suggesting that the meteorological conditions might explain approximately 10 μg $m^{-3}$ of $PM_{2.5}$ variation. The inference here was mainly based on the studies from Ding et al. (2021) and Zhai et al. (2019). Ding et al. (2021) found that the mean measured $PM_{2.5}$ mass concentration and dispersion coefficient normalized $PM_{2.5}$ mass concentration were 67.5 and 45.6 μg $m^{-3}$ during the COVID-19 lockdown period, indicating that unfavorable meteorological condition generated approximately 22 μg $m^{-3}$ growth of $PM_{2.5}$. Zhai et al. (2019) found that the mean $PM_{2.5}$ decrease across China was 4.6 μg $m^{-3}$ $yr^{-1}$ in the meteorology-corrected data from 2013 to 2018, 12% lower than in the original data, meaning that 12% of the $PM_{2.5}$ decrease in the original data was attributable to the meteorology. Furthermore, we also added the related contents from other studies in the revised manuscript (on the lines 422-426).

Ding, J., Dai, Q. L., Li, Y. F., Han, S. Q., Zhang, Y. F., and Feng, Y. C.: Impact of meteorological condition changes on air quality and particulate chemical composition during the COVID-19 lockdown, J. Environ. Sci., 109, 45-56, https://doi.org/10.1016/j.jes.2021.02.022, 2021.
Zhai, S., Jacob, D.J., Wang, X., Shen, L., Li, K., Zhang, Y., 2019. Fine particulate matter ($PM_{2.5}$) trends in China, 2013–2018: separating contributions from anthropogenic emissions and meteorology. Atmos. Chem. Phys. 19, 11031-11041.

7. L542-581, this paper analyzed the changes in source contributions in the winter heating periods

in Qingdao from 2011-2012, 2016, and 2019. The author should clarify the reasons and necessity of this analysis.

**Response:** Thanks for your advice. The related reasons and necessity have been added in the revised manuscript (on the lines 563-569). The more details as following:

Furthermore, with the beginning of heating season in northern cities in China (Liu et al., 2016; Li et al., 2019a), the atmospheric pollutant emissions increased substantially (Chen et al., 2020a). Coupled with the adverse meteorological conditions (Li et al., 2019a), haze episodes occurred frequently during this period (Liu et al., 2018a; Yang et al., 2020). Therefore, the control effects of pollution sources and key control sources in the specific period can be better highlighted through analyzing the changes in the contributions of emission sources during heating seasons over the years.

Chen, J.Y., Shan, M., Xia, J.J., and Jiang, Y.: Effects of space heating on the pollutant emission intensities in "2+26" cities, Building Environ., 175, 106817, https://doi.org/10.1016/j.buildenv.2020.106817, 2020a.

Li, H., You, S.J., Zhang, H., Zheng, W.D., and Zou, L.J.: Analysis of the impacts of heating emissions on the environment and human health in North China, J. Clean Prod., 207, 728-742, https://doi.org/10.1016/j.jclepro.2018.10.013, 2019a.

Liu, B.S., Song, N., Dai, Q.L., Mei, R.b., Sui, B.H., Bi, X.H., and Feng, Y.C.: Chemical composition and source apportionment of ambient $PM_{2.5}$ during the non-heating period in Taian, China, Atmos. Res., 170, 23-33, https://doi.org/10.1016/j.atmosres.2015.11.002, 2016.

Liu, B.S., Cheng, Y., Zhou, M., Liang, D.N., Dai, Q.L., Wang, L., Jin, W., Zhang, L.Z., Ren, Y.B., Zhou, J.B., Dai, C.L., Xu, J., Wang, J., Feng, Y.C., and Zhang, Y.F.: Effectiveness evaluation of temporary emission control action in 2016 in winter in Shijiazhuang, China, Atmos. Chem. Phys., 18, 7019-7039, https://doi.org/10.5194/acp-18-7019-2018, 2018a.

Yang, S., Duan, F., Ma, Y., Li, H., Ma, T., Zhu, L., Huang, T., Kimoto, T., and He, K.: Mixed and intensive haze pollution during the transition period between autumn and winter in Beijing, China, Sci. Total Environ., 711, 134745, https://doi.org/10.1016/j.scitotenv.2019.134745, 2020.

**Comments of Anonymous Referee #1**
**Response:** Thanks for your advice. We have checked your comments carefully and found that your comments have nothing to do with our research. It seems like you have uploaded a wrong file. Therefore, we cannot make relevant modifications.

This manuscript investigated chemical environment for surface $O_3$ for six major industrial regions across China in summer 2016. Detailed chemistry-climate model simulations were employed to diagnose ozone sensitivity to precursors and contrast the effectiveness of different measures to reduce surface $O_3$ concentrations. This manuscript is helpful to understand ozone pollution mechanism in Chinese cities, and within the scope of ACP. I think it is publishable in ACP after my following concerns are addressed.

Line 215: The gross rate of production $P(O_3)$ actually represents the production rate of $O_X$ ($O_3$ + $NO_2$) through the reaction $HO_2$ ($RO_2$) +NO. Therefore, the net ozone production rate should include the loss term $NO_2$+OH (Wang et al., 2019. doi.org/10.5194/acp-19-9413-2019). In addition to

OH+NO$_2$ and RO$_2$+NO$_2$, the loss of NOx should also include RO$_2$+NO and OH+HONO When calculating OPE. Please give specific quantification even though these reactions play a minor role in the loss of NOx.

Figure 4 shows significant underestimation for NO$_2$ in daytime, but overestimation for NO$_2$ at nighttime. The overestimation of NO$_2$ at night maybe related to underestimated nighttime chemistry such as the removal of NO$_3$ and N$_2$O$_5$ through heterogenous uptake (Li et al., 2018; Li et al., 2019). A short discuss should be performed. Additionally, how do these underestimation and overestimation for NO$_2$ influence your diagnosis of ozone sensitivity? For example, the underestimation of NO$_2$ in Chongqing will lead to more NOx-limited, which likely misleads the actual situation.

Figure 8. shows ozone increased from 70 ppb to over 80 ppb during 2013-2019. However, observed ozone concentrations in Beijing didn't increased significantly during the period or decreased after 2015 in spite that ozone increased over North China Plain (Lu et al., 2018. DOI: 10.1021/acs.estlett.8b00366; Tang et al., 2020. doi.org/10.1016/j.atmosres.2020.105333). This needs further explanations.

Line 270: How do you obtain VOC and NOx emissions in 2018 and 2019 given that Cheng et al (2019) just estimated emissions during 2013-2017. Please give specific description.

Line 145: There are only 450 measurement stations in 2013, growing to 1,500 stations in 2017 and 1670 stations in 2019.

Line 300: "summer-mean ozone" should be "daily mean ozone".

References:

Li, J., Chen, X., Wang, Z., Du, H., Yang, W., Sun, Y., Hu, B., Li, J., Wang, W., and Wang, T.: Radiative and heterogeneous chemical effects of aerosols on ozone and inorganic aerosols over East Asia, Science of the Total Environment, 622, 1327-1342, 2018.

Li, K., Jacob, D. J., Liao, H., Zhu, J., Shah, V., Shen, L., Bates, K. H., Zhang, Q., and Zhai, S.: A two-pollutant strategy for improving ozone and particulate air quality in China, Nature Geoscience, 12, 906-910, 10.1038/s41561-019-0464-x, 2019.

---

## Referee Report (RR1)

The manuscript "Dramatic changes in atmospheric pollution source contributions for a coastal megacity in Northern China from 2011 to 2020" by Baoshuang Liu et al. reports the long-term variations of major air pollutants. By applying a Random Forest method, Theil-Sen regression, and dispersion normalization, the authors separated the contribution of meteorology and that of clean air actions to the air pollution mitigation. I am so sorry that I have uploaded a wrong comment file during the first round of the review. After noticed that, I went through both the original and the revised manuscripts. In general, I think they are well written and provide valuable information to the community. I recommend the publication after some minor revisions. Please noted that my following comments are referring to the revised manuscript.

**General comments**

1. The authors essentially applied two methods (i.e., RF and VC) to decouple the influence of meteorology. It would be interested to know to which extent the corrected temporal variations agree to each other. Also, it is better to give a short explanation of using VC instead of RF correction for the analysis in Section 3.3 and 3.4.

2. In section 3.3.2, the authors used VC for correcting the meteorological influence on the source apportionment results. How would this be compared with the source apportionment derived from the VC corrected PM concentrations?

**Specific comments**

**Line 102 − 103, Page 4:** It is better to give a quantitative description (e.g., AQI or PM2.5 changes) on "greatly improved".

**Line 147, Page 6:** Are the sampling instruments home-built or commercial? Please also specify the size of the sampling filter and the sampling flow rate.

**Line 386 − 389, Page 14:** While the enhancement of atmospheric oxidation can certainly cause O3 increase, the strong decrease of NO2 (by almost the same percentage as that of O3 increase) indicating a weakened "NOx titration effect" which may also result in higher O3 levels, especially during cold seasons when photooxidation is usually weak.

**Line 416 − 418, Page 15:** The unit for VC should be "$m^2\,s^{-1}$".

---

## Author Response (AR3)

**Comments of Anonymous Referee #1**

The manuscript "Dramatic changes in atmospheric pollution source contributions for a coastal megacity in Northern China from 2011 to 2020" by Baoshuang Liu et al. reports the long-term variations of major air pollutants. By applying a Random Forest method, Theil-Sen regression, and dispersion normalization, the authors separated the contribution of meteorology and that of clean air actions to the air pollution mitigation. I am so sorry that I have uploaded a wrong comment file during the first round of the review. After noticed that, I went through both the original and the revised manuscripts. In general, I think they are well written and provide valuable information to the community. I recommend the publication after some minor revisions. Please noted that my following comments are referring to the revised manuscript.

General comments

1. The authors essentially applied two methods (i.e., RF and VC) to decouple the influence of meteorology. It would be interested to know to which extent the corrected temporal variations agree to each other. Also, it is better to give a short explanation of using VC instead of RF correction for the analysis in Section 3.3 and 3.4.

**Response:** Thanks for your advice. (1) The RF-based weather normalization method can well decouple the overall weather effects, while the VC-normalization can only decouple the local dispersion. VC-normalization is relatively simple but needs VC measurement data to be known a priori, while RF-based weather normalization needs a large size of data to well training the model before de-weathering. *The fact that there is a big difference in the size and time-resolution between the two datasets (routine air quality data versus PM chemical composition data)*, we therefore chose two methods rather than a "one-size-fits-all" approach to decouple the "weather effects" based on the strengths and limitations of methodologies. To this end, we chose RF-based weather normalization to de-weathering for air quality data that measured in 2015-2020, and used VC-normalization for offline filter-based measured chemical compositional data. A comparison of the two methods sounds desirable but does not make physical sense cause both methods fit their own purposes. (2) In this study, we added related explanation of using VC instead of RF normalization in the revised manuscript (on the lines 203-211). The more details as following:

Although the RF-based weather normalization method can well decouple the overall weather effects, it needs a large size of data to well training the model before de-weathering. The fact that there is a big difference in the size and time-resolution between the routine air quality data and the offline filter-based measured $PM_{2.5}$ chemical compositional data. However, the meteorological dispersion can be quantified by the ventilation coefficient (VC) (Kleinman et al., 1976; Iyer and Raj, 2013). Although the VC-normalization that needs VC data to be known a priori can only decouple the local dispersion, it is relatively simple and useful to decouple the impact of dispersion (Ding et al., 2021). Therefore, this normalized approach is very suitable for the offline data with small size and poor continuity.

2. In section 3.3.2, the authors used VC for correcting the meteorological influence on the source apportionment results. How would this be compared with the source apportionment derived from the VC corrected PM concentrations?

**Response:** Thanks for your advice. The reviewer's suggestion is very good. Source apportionment can be conducted by the PM composition data after dispersion normalization, we have carried out

relevant research in the early stage (as shown in Dai et al. (2020)), and constructed the dispersion normalized PMF (DN-PMF); firstly, we normalized the PM composition data using the VC data during the study period, and then carried out the source apportionment by these normalized data so that the apportioned results can more accurately reflect the impact of emission sources. In contrast, this study mainly used the VC data to normalize the results of source analysis to correct the impact of meteorological conditions, to better reflect the impact of emission sources. In fact, we are carrying out relevant studies on the comparison of the two methods, and the relevant results will be published in the future. However, the related analysis was obviously beyond the scope of this study. The purpose of this study was to use the mature normalized methods to analyze the changes in emission sources in Qingdao in recent 10 years.

Dai, Q. L., Liu, B. S., Bi, X. H., Wu, J. H., Liang, D. N., Zhang, Y. F., Feng, Y. C., and Hopke, P. K.: Dispersion Normalized PMF Provides Insights into the Significant Changes in Source Contributions to $PM_{2.5}$ after the COVID-19 Outbreak, Environ. Sci. Technol., 54, 9917-9927, https://doi.org/10.1021/acs.est.0c02776, 2020.

Specific comments

Line 102 -103, Page 4: It is better to give a quantitative description (e.g., AQI or $PM_{2.5}$ changes) on "greatly improved".

**Response:** Thanks for your advice. We added the quantitative descriptions on "greatly improved" in the revised manuscript (on the lines 101-103). The more details as following:

Up to now, the air quality in Qingdao has been greatly improved, the annual mean concentrations of $PM_{2.5}$ and $PM_{10}$ all decreased by 38% from 2015 to 2020 based on the air quality monitoring data.

Line 147, Page 6: Are the sampling instruments home-built or commercial? Please also specify the size of the sampling filter and the sampling flow rate.

**Response:** Thanks for your advice. The sampling instruments are commercial and the more details on the instrument corporations, sampling filter and the sampling flow rate are shown in Table S3.

**Table S3.** Details of sampling instruments and filters during different sampling years.

| Year | Instrument | Model | Corporation | Country | Flow rate (L $min^{-1}$) | Filter diameter (mm) | Filter category | Corporation | Country |
|------|-----------|-------|-------------|---------|--------------------------|----------------------|-----------------|-------------|---------|
| 2011-2012 | Four channel air particulate matter sampler | TH-16A | Wuhan Tianhong Instrument Co., Ltd | China | 16.7 | 47 | Polypropylene/ Quartz | Beijing Synthetic Fiber Research Institute/Pall Life Sciences | China/ USA |
| 2016 | Multichannel ambient air particulate sampler | ZR-3930D | Qingdao Junray Intelligent Instrument Co., Ltd | China | 16.7 | 47 | Polypropylene/ Quartz | Munktell | Sweden |

| 2019 | Multichannel ambient air particulate sampler | ZR-3930D | Qingdao Junray Intelligent Instrument Co., Ltd | China | 16.7 | 47 | Polypropylene/ Quartz | Pall Life Sciences | USA |

Line 386-389, Page 14: While the enhancement of atmospheric oxidation can certainly cause $O_3$ increase, the strong decrease of $NO_2$ (by almost the same percentage as that of O3 increase) indicating a weakened "NOx titration effect" which may also result in higher $O_3$ levels, especially during cold seasons when photooxidation is usually weak.

**Response:** Thanks for your advice. We very agree with the reviewer's suggestion. The related explanations have been added in the revised manuscript (on the lines 386-388). The more details as following:

Meanwhile, the markedly decrease of $NO_2$ during the full lockdown can also weaken "NOx titration effect", further resulting in higher $O_3$ level during this period.

Line 416-418, Page 15: The unit for VC should be "$m^2\ s^{-1}$"

**Response:** Thanks for your advice. The unit of VC has been modified in the revised manuscript (on the lines 414 and 416).

**Comments of Anonymous Referee #2**

The manuscript applied a machine learning-based meteorological normalization approach to decouple the meteorological effects from air quality trend in a coastal city in northern China, and further assessed the changes in the contributions of pollution sources in Qingdao in the past ten years. There are some minor issues need to be solved.

1. In the last paragraph of the introduction, if any, it is suggested to add some evaluation articles on the changes in air quality and the effectiveness of control measures before and after major events held in Qingdao (e.g., the 2018 the Shanghai Cooperation Organization summit in Qingdao). Clarify the differences with this paper to support the particularity of this study.

**Response:** Thanks for your advice. We added the related contents in the revised version (on the lines 103-107). The more details as following:

Liu et al. (2020a) assessed the changes in $O_3$ concentrations during the Shanghai Cooperation Organization (SCO) Summit in Qingdao and analyzed the impact of control measures on the emissions reduction of its precursors, and Liu et al. (2020b) also analyzed the reasons for the increase of $O_3$ concentration at nighttime during the SCO Summit.

2. In the sampling and analysis section (L144-145), what is the basis for collecting 22 hours a day? Why not 24 hours?

**Response:** Thanks for your advice. The sampling period was from 11:00 to 09:00 the next day, and the sampling time was 22 hours. The sampling time covers the peak time of morning and evening traffic and the period of strong photochemical reaction at noon, and includes the peaks and troughs of air pollutant concentration. The sampling period of this study is representative. Meanwhile,

because the sampling instruments in this study cannot continuously replace the filters, therefore, it needs to take a certain time to replace the sampling filters for subsequent sampling. Furthermore, similar sampling times (22-23 hours) occurred in the studies of Liu et al., (2021), Fang et al. (2017), Turap et al. (2019), and Wang et al. (2016).

Liu, B.S., Wu, J.H., Wang, J., Shi, L.Y., Meng, H., Dai, Q.L., Wang, J., Song, C.B., Zhang, Y.F., Feng, Y.C., Hopke, P.K., 2021. Chemical characteristics and sources of ambient $PM_{2.5}$ in a harbor area: Quantification of health risks to workers from source-specific selected toxic elements. Environ. Pollut., 2021, 268, 115926.

Fang, X.Z., Bi, X.H., Xu, H., Wu, J.H., Zhang, Y.F., Feng, Y.C., 2017. Source apportionment of ambient $PM_{10}$ and $PM_{2.5}$ in Haikou, China. Atmos. Res. 190, 1–9.

Turap, Y., Talifu, D., Wang, X.M., Abulizi, A., Maihemuti, M., Tursun, Y., Ding, X., Aierken, T., Rekefu, S., 2019. Temporal distribution and source apportionment of $PM_{2.5}$ chemical composition in Xinjiang, NW-China. Atmos. Res. 218, 257–268.

Wang, Y.N., Jia, C.H., Tao, J., Zhang, L.M., Liang, X.X., Ma, J.M., Gao, H., Huang, T., Zhang, K., 2016. Chemical characterization and source apportionment of $PM_{2.5}$ in a semi-arid and petrochemical-industrialized city, Northwest China. Sci. Total Environ. 573, 1031–1040.

3. Have the sample data of 2011-2012, 2016, and 2019 been carried out for source analysis, respectively? Please clarify. In addition, the data of all sites were chronologically ordered end to end for each PMF analysis? PMF is usually used for source analysis based on the long-time data of one site. Please explain the rationality.

**Response:** (1) Yes. The related contents were described in the lines 475-477. (2) Yes. The data of all sites were chronologically ordered end to end for each PMF analysis, and the related contents have shown in text S2 in the supplementary material. (3) In this study, the data of all sampling sites were included in PMF for source analysis, mainly because all sampling sites were located in Qingdao urban area, and the impact source-categories at these sites were the same. The stability of the model running can be improved due to the increase of the number of samples by incorporating multiple sites data into PMF. Escrig et al. (2009) suggested that because the three sampling sites were affected by the same source-categories, therefore, they can be treated as one site and then included in PMF for analysis. Mooibroek et al. (2011) incorporated the receptor data of five sampling sites into PMF for source apportionment, which considered that combining the receptor data of five sites can increase the amount of input data for the model, and then more stable analysis results can be obtained. Furthermore, many related studies have adopted similar methods (Lu et al., 2018; Ezeh et al., 2019; Zhao et al., 2019).

Escrig, A., Monfort, E., Celades, I., Querol, X., Amato, F., Minguillón, M.C., and Hopke, P.K.: Application of optimally scaled target factor analysis for assessing source contribution of ambient $PM_{10}$, J. Air & Waste Manage., 59, 1296–1307, https://doi.org/10.3155/1047-3289.59.11.1296, 2009.

Mooibroek, D., Schaap, M., Weijers, E.P., and Hoogerbrugge, R.: Source apportionment and spatial variability of $PM_{2.5}$ using measurements at five sites in the Netherlands, Atmos. Environ., 45, 4180-4191, https://doi.org/10.1016/j.atmosenv.2011.05.017, 2011.

Lu, Z.J., Liu, Q. Y., Xiong, Y., Huang, F., Zhou, J. B., and Schauer, J.J.: A hybrid source apportionment strategy using positive matrix factorization (PMF) and molecular marker chemical mass balance (MM-CMB) models, Environ Pollut., 238, https://doi.org/10.1016/j.envpol.2018.02.091, 2018.

Ezeh, G, C., Obioh, I, B., Asubiojo, O., Abiye, O, E., and Onyeuwaoma, N, D.: A study of $PM_{2.5-10}$ pollution at three functional receptor sites in a sub-Saharan African megacity, Aerosol Science and Engineering., 3, https://doi.org/10.1007/s41810-019-00044-3, 2019.

Zhao, Z.P., Lv, S., Zhang, Y.H., Zhao, Q.B., Shen, L., Xu, S., Yu, J.Q., Hou, J.W., and Jin, C.Y.: Characteristics and source apportionment of $PM_{2.5}$ in Jiaxing, China, Environ. Sci. Pollut. R., 26, https://doi.org/10.1007/s11356-019-04205-2, 2019.

4. In the process of PMF calculation, the author analyzed DISP and BS (Fig. S9-11, Table S14), but the author did not mention it in the introduction of PMF method. Please add.

**Response:** Thanks for your advice. The related contents have been added in the revised manuscript (on the lines 239-240). The more details as following:

Bootstrap (BS) and displacement (DISP) analyses were used to investigate the effects of measurement error and rotation ambiguity on the resulting solutions.

5. In recent years, scholars have studied the change of air quality in many cities around the world (such as Beijing), many of which use random forest and other methods. In section 3.11, it is suggested to add the content of comparative analysis with other cities.

**Response:** Thanks for your advice. We added the contents of comparative analysis with other cities in the revised manuscript (on the lines 288-299). The more details as following:

Similar to this study, Vu et al. (2019) found that primary emission controls required by the CAAP in Beijing have led to substantial reductions in $PM_{2.5}$, $PM_{10}$, $NO_2$, $SO_2$, and CO from 2013 to 2017 of approximately 34%, 24%, 17%, 68%, and 33%, respectively, after meteorological normalization. Zhai et al. (2019) suggested that the mean $PM_{2.5}$ decreased across China was 4.6 µg $m^{-3}$ $yr^{-1}$ in the meteorology-corrected data from 2013 to 2018, and the Beijing–Tianjin–Hebei, the Yangtze River Delta, the Pearl River Delta, the Sichuan Basin, and the Fenwei Plain decreased 8.0, 6.3, 2.2, 4.9, and 5.0 µg $m^{-3}$ $yr^{-1}$, respectively. Overall, the concentrations of most air pollutants (i.e., $PM_{2.5}$, $PM_{10}$, $SO_2$, $NO_2$, and CO) in China have showed a decreasing trend in recent years (Zhao et al., 2021; Fan et al., 2020, while that of $O_3$ has performed an increasing trend (Li et al., 2020; Ma et al., 2021), which further facilitated secondary particles formation (Nøjgaard et al., 2012).

6. L401-404 "The observed and normalized concentrations of $PM_{2.5}$ during the whole study period were 93 and 83 µg $m^{-3}$, suggesting that unfavorable meteorological conditions generated approximately 10 µg $m^{-3}$ of growth of $PM_{2.5}$", can the difference of simple subtraction represent the influence value of meteorology? Is there any basis?

**Response:** Thanks for your advice. In this study, we used the ventilation coefficient to normalize the impact on $PM_{2.5}$ concentrations from the meteorology, and the observed and normalized concentrations of $PM_{2.5}$ during the whole study period were 93 and 83 µg $m^{-3}$, respectively,

suggesting that the meteorological conditions might explain approximately 10 μg m$^{-3}$ of PM$_{2.5}$ variation. The inference here was mainly based on the studies from Ding et al. (2021) and Zhai et al. (2019). Ding et al. (2021) found that the mean measured PM$_{2.5}$ mass concentration and dispersion coefficient normalized PM$_{2.5}$ mass concentration were 67.5 and 45.6 μg m$^{-3}$ during the COVID-19 lockdown period, indicating that unfavorable meteorological condition generated approximately 22 μg m$^{-3}$ growth of PM$_{2.5}$. Zhai et al. (2019) found that the mean PM$_{2.5}$ decrease across China was 4.6 μg m$^{-3}$ yr$^{-1}$ in the meteorology-corrected data from 2013 to 2018, 12% lower than in the original data, meaning that 12% of the PM$_{2.5}$ decrease in the original data was attributable to the meteorology. Furthermore, we also added the related contents from other studies in the revised manuscript (on the lines 421-425).

Ding, J., Dai, Q. L., Li, Y. F., Han, S. Q., Zhang, Y. F., and Feng, Y. C.: Impact of meteorological condition changes on air quality and particulate chemical composition during the COVID-19 lockdown, J. Environ. Sci., 109, 45-56, https://doi.org/10.1016/j.jes.2021.02.022, 2021.
Zhai, S., Jacob, D.J., Wang, X., Shen, L., Li, K., Zhang, Y., 2019. Fine particulate matter (PM$_{2.5}$) trends in China, 2013–2018: separating contributions from anthropogenic emissions and meteorology. Atmos. Chem. Phys. 19, 11031-11041.

7. L542-581, this paper analyzed the changes in source contributions in the winter heating periods in Qingdao from 2011-2012, 2016, and 2019. The author should clarify the reasons and necessity of this analysis.

**Response:** Thanks for your advice. The related reasons and necessity have been added in the revised manuscript (on the lines 560-566). The more details as following:
Furthermore, with the beginning of heating season in northern cities in China (Liu et al., 2016; Li et al., 2019a), the atmospheric pollutant emissions increased substantially (Chen et al., 2020a). Coupled with the adverse meteorological conditions (Li et al., 2019a), haze episodes occurred frequently during this period (Liu et al., 2018a; Yang et al., 2020). Therefore, the control effects of pollution sources and key control sources in the specific period can be better highlighted through analyzing the changes in the contributions of emission sources during heating seasons over the years.

Chen, J.Y., Shan, M., Xia, J.J., and Jiang, Y.: Effects of space heating on the pollutant emission intensities in "2+26" cities, Building Environ., 175, 106817, https://doi.org/10.1016/j.buildenv.2020.106817, 2020a.
Li, H., You, S.J., Zhang, H., Zheng, W.D., and Zou, L.J.: Analysis of the impacts of heating emissions on the environment and human health in North China, J. Clean Prod., 207, 728-742, https://doi.org/10.1016/j.jclepro.2018.10.013, 2019a.
Liu, B.S., Song, N., Dai, Q.L., Mei, R.b., Sui, B.H., Bi, X.H., and Feng, Y.C.: Chemical composition and source apportionment of ambient PM$_{2.5}$ during the non-heating period in Taian, China, Atmos. Res., 170, 23-33, https://doi.org/10.1016/j.atmosres.2015.11.002, 2016.
Liu, B.S., Cheng, Y., Zhou, M., Liang, D.N., Dai, Q.L., Wang, L., Jin, W., Zhang, L.Z., Ren, Y.B., Zhou, J.B., Dai, C.L., Xu, J., Wang, J., Feng, Y.C., and Zhang, Y.F.: Effectiveness evaluation of temporary emission control action in 2016 in winter in Shijiazhuang, China, Atmos. Chem. Phys., 18, 7019-7039, https://doi.org/10.5194/acp-18-7019-2018, 2018a.

Yang, S., Duan, F., Ma, Y., Li, H., Ma, T., Zhu, L., Huang, T., Kimoto, T., and He, K.: Mixed and intensive haze pollution during the transition period between autumn and winter in Beijing, China, Sci. Total Environ., 711, 134745, https://doi.org/10.1016/j.scitotenv.2019.134745, 2020.